# Development and Optimization of the Biological Conversion of Ethane to Ethanol Using Whole-Cell Methanotrophs Possessing Methane Monooxygenase

**DOI:** 10.3390/molecules24030591

**Published:** 2019-02-07

**Authors:** So Hyeon Oh, In Yeub Hwang, Ok Kyung Lee, Wangyun Won, Eun Yeol Lee

**Affiliations:** 1Department of Chemical Engineering, Kyung Hee University, Gyeonggi-do 17104, Korea; ossoyam@naver.com (S.H.O.); francis2@khu.ac.kr (I.Y.H.); okblessyou@khu.ac.kr (O.K.L.); 2Department of Chemical Engineering, Changwon University, Changwon 51140, Korea; wwon@changwon.ac.kr

**Keywords:** methane monooxygenase, methanotrophs, ethane, ethanol, gas-to-liquid

## Abstract

The biological production of ethanol from ethane for the utilization of ethane in natural gas was investigated under ambient conditions using whole-cell methanotrophs possessing methane monooxygenase. Several independent variables including ethane concentration and biocatalyst amounts, among other factors, were optimized for the enhancement of ethane-to-ethanol bioconversion. We obtained 0.4 g/L/h of volumetric productivity and 0.52 g/L of maximum titer in optimum batch reaction conditions. In this study, we demonstrate that the biological gas-to-liquid conversion of ethane to ethanol has potent technical feasibility as a new application of ethane gas.

## 1. Introduction

Bioethanol is used for chemical feedstock and gasoline blending [1]. More than 20% bioethanol is expected to be added to fuel for motor vehicles in South America and Brazil in order to replace gasoline. Bioethanol-blended gasoline for transportation can reduce both the dependency on petroleum-based fuels and the concern about global warming [2]. Further, ethanol has a higher octane rating than gasoline, allowing for the efficient production of power due to an enhanced compression ratio [3]. On the basis of these advantages and considering the increase in energy consumption associated with population growth and industrial development, the demand for bioethanol is expected to surpass 125 billion liters by 2020, from 46 billion liters in 2007 [2].

The conventional processes for ethanol production are hydration of ethylene and bioethanol fermentation of sugar using yeast. The direct hydration of ethylene is accomplished at 250–300 °C and 70–80 atm in the presence of steam [4]. The requirement of high temperature and high pressure causes an increase in production cost [5]. In the case of bioethanol fermentation, one molecule of glucose produces two molecules of bioethanol with release of two molecules of CO_2_. This causes technical problems such as carbon loss and the generation of greenhouse gases. Another issue of bioethanol fermentation from corn and sugarcane is the problem of increasing food prices by using food resources as an energy source [6]. Thus, lignocellulosic biomass can mitigate food dependency in bioethanol production. However, pretreatment, which is an expensive and complicated process, together with an enzymatic saccharification step, is required to produce bioethanol by fermentation [1,7].

Natural gas composed of methane and ethane remains an untapped energy source or chemical feedstock. Methanotrophs can utilize methane as a sole carbon and energy source, and the first step in their methane utilization is the oxidation of methane by methane monooxygenase (MMO) [8]. Metabolic engineering of methanotrophs for producing value-added products from methane has been conducted, as methane remains a remarkably cheap feedstock [5]. There are two types of MMOs, soluble MMO (sMMO) and particulate MMO (pMMO). pMMO can oxidize short-chain alkanes (generally less than five carbon long) and alkenes, whereas sMMO has a relatively broad substrate specificity and can oxidize aromatics, chlorinated aromatics, as well as alkanes and alkenes [9]. At low concentrations of copper, sMMO is expressed and catalyzes methane oxidation using nicotinamide adenine dinucleotide (NADH) as the reducing agent in the cytosol. At a high level of copper concentration, pMMO is expressed in the intracytoplasmic membranes and uses a reduced ubiquinone as the electron donor [10,11,12]. Some methanotrophs possess both sMMO and pMMO, and the expression of sMMO and pMMO is regulated by copper concentration. Cells expressing pMMO have a higher growth rate due to the higher affinity of pMMO for methane [13,14].

Ethane is used for the production of ethylene by steam cracking. The steam cracking process is one of the most energy-intensive processes in the chemical industry and it causes almost 180–200 million tons of carbon dioxide emissions in the world annually [15]. Recently, the biological oxidation of short-chain alkane gases, such as ethane to liquid alcohols, has been investigated using an engineered cytochrome P450 [16,17]. Ethane can be utilized for ethanol production by methanotrophic bacteria because MMO oxidizes ethane as well as methane because of its broad substrate specificity [9]. This process is not accompanied by carbon dioxide emission since MMO converts one molecule of ethane into one molecule of ethanol in mild conditions, which is an environmentally friendly approach compared to the chemical process [10,18].

Methanol dehydrogenase (MDH) exists in methanotrophs and catalyzes the oxidation of both methanol to formaldehyde and ethanol to acetaldehyde [19]. In methanotrophic whole cells, ethanol can also be consumed by alcohol dehydrogenase [20]. Thus, the inhibition of MDH by chemical inhibitors is required for methanol accumulation. However, the addition of an MDH inhibitor blocks the generation of NADH from the oxidation of methanol to formaldehyde. In order to solve this problem, sodium formate was added to replenish NADH [21]. Whole-cell conversions of ethane to ethanol have been studied using methanotrophs as a biocatalyst. Previous studies have shown that ethane could be converted into ethanol on the basis of MDH inhibition and the addition of sodium formate, which demonstrates conceptually the biotransformation of ethane gas substrate [20,22].

In this study, we compared some representative methanotrophs in order to evaluate their ability of bioconversion of ethane to ethanol and investigated the optimal conditions for obtaining a high titer of ethanol using various concentrations of MDH inhibitor and formate in a quantitative way. In addition, the concentration of the cells and substrate and the pH values were optimized.

## 2. Results and Discussion

### 2.1. Evaluation of Ethane-Oxidizing Bacteria

First, we evaluated methanotrophs for their ability to oxidize ethane to ethanol in batch systems to select the best strain. *Methylomonas* sp. DH-1 and *Methylomicrobium alcaliphilum* 20Z in Type I methanotrophs and *Methylosinus trichosporium* OB3b in type II methanotrophs were tested as the biocatalyst for ethane oxidation.

*Methylomonas* sp. DH-1 and *M. alcaliphilum* 20Z have pMMO. *M. trichosporium* OB3b has both sMMO and pMMO. When *M. trichosporium* OB3b was cultured in a growth medium containing 5 μM CuSO_4_, pMMO was expressed, and the expression of sMMO was inhibited, increasing cell growth [23]. All bioconversion reactions in this study were conducted on the basis of pMMO activity, which has a higher affinity for short-chain alkanes than sMMO [14]. Quantitative analysis of the samples revealed that *M. trichosporium* OB3b produced more ethanol than other strains, and *M. alcaliphilum* 20Z did not accumulate ethanol from ethane (Figure 1). In a previous study, whole-cell experiments with strain 20Z showed pH-dependent rates of methane oxidation, with the highest rate at pH 9.0 and a much slower rate at pH 7.0 or 10.0 [24]. Thus, *M. alcaliphilum* 20Z showed the lowest ethanol production among three strains at pH 7.0. The maximum ethanol concentration of 0.476 g/L was obtained by *M. trichosporium* OB3b, 4.25 times higher than the ethanol concentration of 0.112 g/L produced by *Methylomonas* sp. DH-1. Thus, we concluded that *M. trichosporium* OB3b was more suitable as a biocatalyst in ethane-to-ethanol production under the conditions analyzed.

### 2.2. Optimization of the Reaction Parameters in Batch Ethanol Production

The effect of ethane concentration on ethanol production was investigated using 0.6 g DCW/L of *M. trichosporium* OB3b whole cells as the biocatalyst in 20 mM sodium phosphate buffer (pH 7) (Figure 2a). One mol of alkane and one mol of O_2_ participate in alcohol formation by pMMO. We expected that ethanol production would decrease when more than 20% (*v*/*v*) of ethane was injected, because approximately 17.4% (*v*/*v*) of ethane and 82.6% (*v*/*v*) of air were needed for maximum ethanol production, theoretically. However, the highest ethanol production was performed at 30% (*v*/*v*) of ethane concentration, and ethane supply of more than 0.288 mmol did not lead to an increase in ethanol accumulation. Excessive amounts of ethane greater than the theoretical value were required for maximum ethanol production. Similar results have been observed in previous experiments of methane-to-methanol conversion using *M. trichosporium* OB3b [23], *Methylocella tundrae* [8], and *Methylomonas* sp. DH-1 [25], in which the maximum methanol concentration was obtained when the supplied methane was more than 17% (*v*/*v*). This result was probably due to the low solubility of methane [25]. The water solubility of methane, ethane, and dioxygen at 20 °C and 1 atm are 0.023 g/kg, 0.06 g/kg, and 0.045 g/kg, respectively, which correspond to 1.44 mM, 2.00 mM, and 1.41 mM, respectively [26].

The effect of pH on ethanol production in the presence of 30% (*v*/*v*) ethane is shown in Figure 2b. The ethanol concentration was the highest at pH 6 and sharply decreased above pH 7, indicating that MMO activity might be inhibited especially under alkaline conditions. In the previous studies of methane-to-methanol conversion by MMO, the optimal pH was around 7 [8,25,27,28].

In order to assess the effect of biocatalyst concentration on ethanol formation, various cell concentrations from 0.15 to 4.8 g DCW/L were examined (Figure 2c). Ethanol formation increased with an increase in cell concentrations up to 2.4 g DCW/L and then decreased at 4.8 g DCW/L. While a higher biocatalyst loading brought rapid ethane conversion to ethanol, excessive biocatalyst loading over the optimum amount led to the decline of ethanol productivity. One possible explanation of this result would be that the amount of MDH inhibitor for 4.8 g/L cell concentration was not enough to completely inhibit the total activity of MDH in the given reaction condition. With respect to the economic aspects, the cell amount needs to be minimized. Thus, the optimal cell loading for ethane-to-ethanol conversion was determined to be 2.4 g/L in terms of cost and production efficiency.

In order to investigate the effect of sodium formate concentration on ethane-to-ethanol conversion, reactions were conducted with 2.4 g DCW/L of methanotrophic cells with various concentrations of sodium formate (Figure 3). The replenishment of NADH by supplying sodium formate increased the ethanol titer in ethane-to-ethanol conversion by transferring electrons to the MMO [21]. The formation of ethanol was enhanced as the formate concentration increased until 80 mM. However, formate concentration above 80 mM was not profitable for ethane oxidation and resulted in a decrease of ethanol concentration. This was probably due to the high concentration of Na-formate that causes the disruptions of intracytoplasmic membranes where pMMO binds [21]. An 80 mM formate was chosen for maximum ethanol production.

In order to accumulate ethanol, further oxidation of ethanol should be prevented by the inhibition of MDH. Methanol was reported to be a competitive inhibitor of ethanol on MDH, but ethanol was not efficiently accumulated by using methanol as an MDH inhibitor [20]. EDTA, phosphate, MgCl_2_, NaCl, and NH_4_Cl have been used as MDH inhibitors in recent studies. Studies on methanol production using methanotrophic bacteria from methane showed that EDTA and phosphate demonstrate a relatively higher MDH inhibition effect than other inhibitors (MgCl_2_, NaCl, NH_4_Cl) [27]. On the basis of these results, EDTA and phosphate were used as MDH inhibitors in this study. The effect of the EDTA and phosphate concentration on ethanol accumulation was examined with 0–10 mM EDTA and 0–800 mM potassium phosphate buffer. Maximum ethanol concentrations of 0.33 and 0.45 g/L were obtained in the presence of 2.4 g DCW/L with the supply of 300 mM phosphate and 0.3 mM EDTA, respectively (Figure 4). Ethanol accumulation with the addition of 0.3 mM EDTA was 1.35 times higher than with 300 mM phosphate. A further increase of inhibitors over 0.3 mM EDTA or 300 mM phosphate resulted in a decrease of ethanol concentration. Cytochrome c_L_ is the primary electron acceptor of MDH, and the electron is expected to be transferred by the coupling of lysyl residues on MDH surface and carboxyl groups of cytochrome c_L_. EDTA hinders the cytochrome from moving to the optimal position for the next electron transfer as it binds to MDH, which inhibits MDH activity [29]. In addition, EDTA can play a role in separating lipopolysaccharide and protein by segregating metal ions from the outer membrane, which results in membrane destruction [30]. Thus, pMMO as a membrane-associated enzyme, can be dissociated and deactivated by a high concentration of EDTA. This is why concentrations above 0.3 mM EDTA reduce ethanol productivity. The decrease of ethanol concentration in the presence of more than 300 mM phosphate could be due to the depression of pMMO catalytic activity by intolerable phosphate concentrations [31]. The production of ethanol from ethane would be practiced on an enormous scale. The use of 300 mM phosphate as a means of retarding ethanol oxidation is not practical because of limited global phosphate reserves and environmental contamination [32]. We concluded that the addition of 0.3 mM EDTA was the most suitable alternative for efficient ethanol accumulation.

To investigate the effect of the cell growth phase of methanotrophic cells used as biocatalysts on ethanol production, *M. trichosporium* OB3b was studied first (Figure 5a). When the initial inoculation concentration was OD_600_ of 0.07, the early exponential phase was started at OD_600_ of 0.6, the stationary phase at OD_600_ of 1.99 was reached after incubation for 33 h, and the OD_600_ value did not change significantly for 11 days. The amount of ethanol was measured when the concentration of the cells used as the biocatalyst corresponded to OD_600_ of 0.6, 1.0, 1.5, and 1.8 (between the exponential phase and the deceleration phase). As shown in Figure 5b, cells in different growth phases between 1.0 and 1.8 did not lead to a significant change in ethanol production, and the maximum concentration of ethanol was obtained using cells harvested at OD_600_ of 1.8. However, the highest conversion efficiency after 1 h of reaction was obtained using cells harvested at OD_600_ of 1.0, which produced ethanol at a concentration more than two times higher than that obtained using cells harvested at OD_600_ of 1.8. This result demonstrates that the cells in the middle exponential phase were more suitable for use as catalysts in ethanol conversion from ethane, since they showed higher efficiency than cells in other phases.

### 2.3. Ethane-to-Ethanol Conversion under Optimized Batch Conditions

The bioconversion of ethane to ethanol under optimized conditions was compared with that of the experiment conducted in the absence of 0.3 mM EDTA and 80 mM formate (Figure 6). When both EDTA and formate were not added, ethanol was not formed. The addition of formate accelerated ethanol production, and the supply of MDH inhibitors such as phosphate and EDTA enhanced ethanol accumulation. The maximum ethanol titer of 0.52 g/L and volumetric productivity of 0.4 g/L/h were achieved from ethane using 2.4 g DCW/L methanotrophic resting cell with 0.3 mM EDTA and 80 mM formate in the batch reaction. These results are almost eight times and six times higher than those previously reported (approximately 0.067 g/L and 0.062 g/L/h) [20].

On the basis of thermodynamic equilibrium analysis with the assumption that MDH was completely inhibited, the maximum ethanol titer was expected to be 1.1 g/L under the optimal conditions. However, the actual ethanol concentration was about 0.52 g/L after 12 h of reaction, because the subsequent oxidation of ethanol into acetaldehyde affected the equilibrium (Figure 6). Considering the consumed amount of ethane, 1.17 g/L ethanol should be accumulated. Under the optimal reaction conditions, 0.52 g/L ethanol, 0.44 g/L acetaldehyde, and 0.01 g/L acetate were accumulated after 12 h due to the incomplete inhibition of MDH (Figure 6). For complete inhibition of MDH, we can consider the option of increasing the MDH inhibitor. However, a high MDH inhibitor concentration can have a negative effect on MMO activity as described above (Figure 4a).

Although the yield of ethane-to-ethanol bioconversion was rather lower than expected, wild-type whole-cell methanotrophs possessing pMMO were successfully used for the bioconversion of ethane gas to liquid ethanol. This study demonstrates that methanotrophs possessing MMO activity could be used as biocatalysts for the biological conversion of ethane to ethanol, as an example of biological gas-to-liquid technology.

## 3. Materials and Methods

### 3.1. Bacterial Strains and Culture Conditions

*M. trichosporium* OB3b, *Methylomonas* sp. DH-1, and *M. alcaliphilum* 20Z were utilized for ethanol production from ethane as biocatalysts. Each cell type was cultured in 500 mL baffled flasks containing 30% (*v*/*v*) methane and 50 mL of nitrate mineral salt (NMS) medium for *Methylomonas* sp. DH-1 [25], *M. trichosporium* OB3b (containing 5 μM CuSO_4_) [23], and *M. alcaliphilum* 20Z [33], respectively. The organisms were cultured in a shaking incubator at 230 rpm and 30 °C.

### 3.2. Conversion of Ethanol from Ethane

The ability of methanotrophs to produce ethanol from ethane was evaluated in 20 mL serum bottles. The grown cells were harvested by centrifugation at 4 °C, 9000× *g* for 20 min. The pelleted cells were rinsed twice with distilled water and reaction buffer (20 mM potassium phosphate). Re-suspended whole cells (0.6 g DCW/L) with the reaction buffer were used as biocatalyst to convert ethane to ethanol. Solutions containing 40 mM formate and 0.5 mM EDTA were added to the reaction buffer in a reaction bottle. The bottle was sealed with a gray butyl rubber septum and an aluminum seal. The batch reactions were initiated by injecting 30% (*v*/*v*) ethane in the headspace and carried out in a shaking incubator at 230 rpm and 30 °C before being terminated by deactivating the biocatalysts using a heating block at 90 °C for 30 min. Subsequently, the concentrations of ethane in the headspace of the cooled samples and of ethanol in the reaction mixture were analyzed. All reactions were done in triplicate.

### 3.3. Analytical Method

Ethane and ethanol were quantified by gas chromatography using a HP-Plot Q capillary column and a flame ionization detector (FID). For ethane analysis, the oven temperature was maintained at 100 °C for 1 min and then raised to 150 °C at a rate of 10 °C/min. The injector and detector temperatures were set at 250 °C and 230 °C, respectively. Nitrogen was used as a carrier gas and flowed at a rate of 2.0 mL/min. For ethanol quantification, the oven temperature was maintained at 200 °C. The injector and detector temperatures were set at 250 °C. The carrier gas flowed at a rate of 2.5 mL/min. Acetaldehyde and acetate were analyzed by HPLC with an Aminex HPX-87H column and DA detector. Acetaldehyde and acetate were detected at 270 nm and 210 nm wavelengths, respectively. The mobile phase was 5 mM H_2_SO_4_.

## 4. Conclusions

Methanotrophs have been considered as promising biocatalysts for the conversion of gaseous alkane to liquid alcohol. In this study, we exploited them as the biocatalysts for ethane-to-ethanol conversion. The batch bioconversion conditions for ethanol production were investigated and optimized. As a result, the optimal conditions were obtained when using 2.4 g/L biocatalyst, 30% (*v*/*v*) ethane, 80 mM sodium formate, and 0.3 mM EDTA. Bioethanol under the optimal conditions was produced with the maximum titer of 0.52 g/L and volumetric productivity of 0.4 g/L/h, respectively. Ethane was successfully converted to ethanol, without the generation of carbon dioxide, simply by employing methanotrophs as the biocatalyst and ethane as the substrate. This study demonstrates that methanotroph-catalyzed bioconversion can be applied for ethanol production from ethane as a biological gas-to-liquid technology.

## Figures and Tables

**Figure 1 molecules-24-00591-f001:**
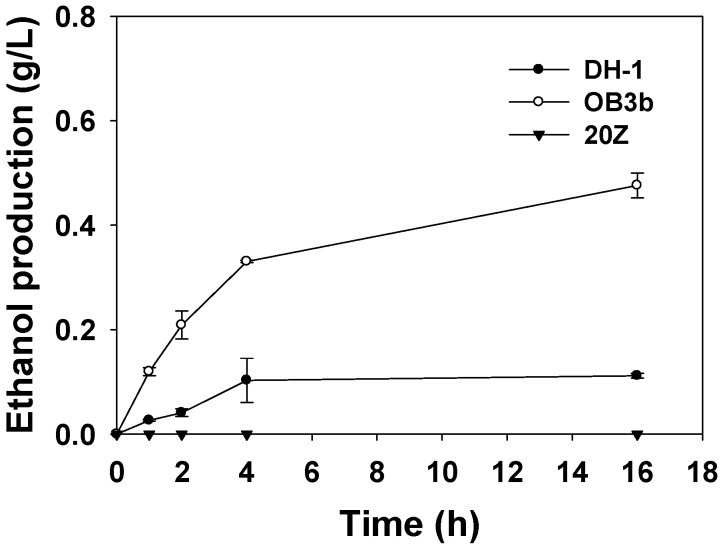
Time course of ethanol production using various methanotrophs.

**Figure 2 molecules-24-00591-f002:**
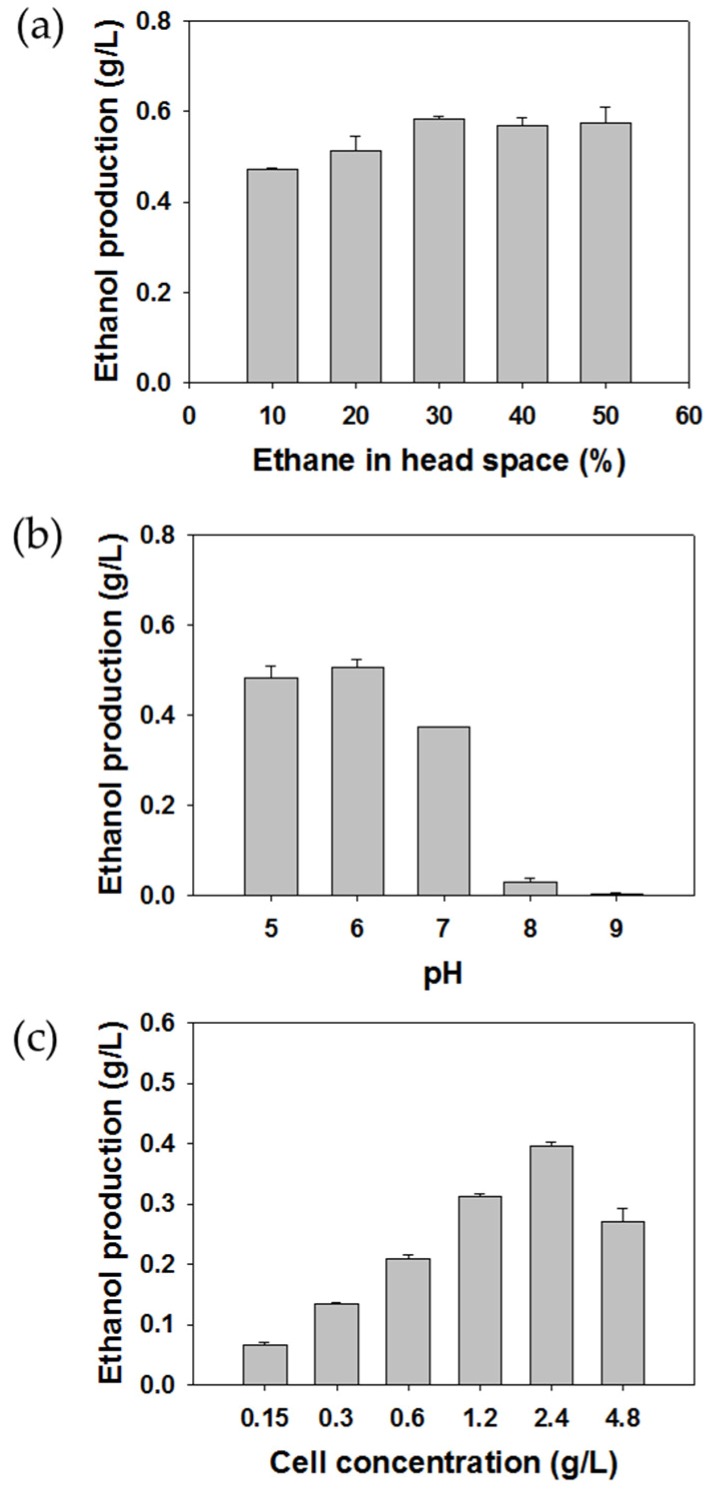
Optimization of ethane-to-ethanol batch conversion parameters such as ethane concentration (**a**), pH value (**b**), inoculum concentration (**c**). The reactions were conducted for 13 h (**a**), 12 h (**b**), and 2 h (**c**) using *Methylosinus trichosporium* OB3b resting cells.

**Figure 3 molecules-24-00591-f003:**
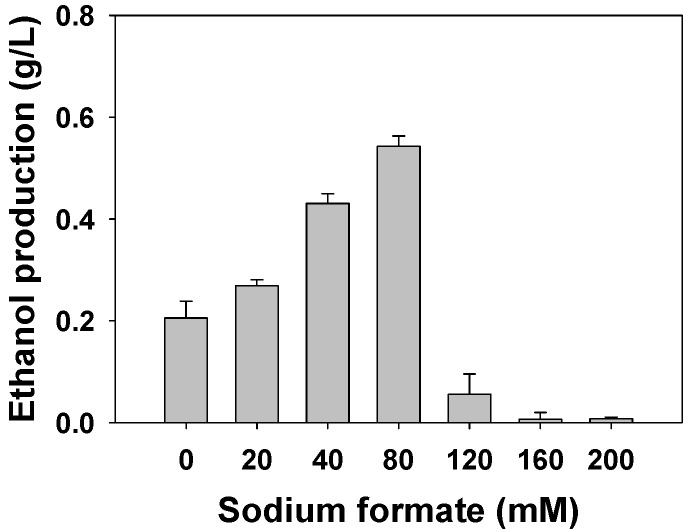
Effect of sodium formate on the oxidation of ethane to ethanol for 12 h.

**Figure 4 molecules-24-00591-f004:**
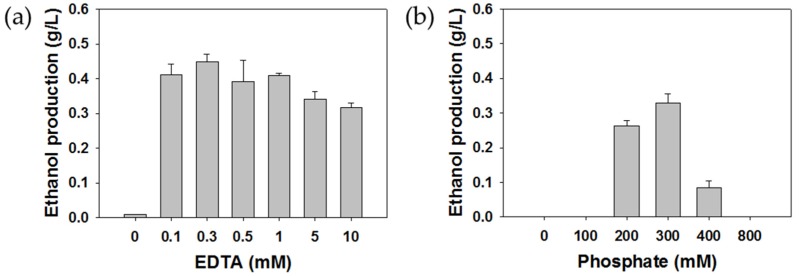
Effects of the addition of EDTA (**a**) and phosphate (**b**) as MDH inhibitors on ethanol accumulation.

**Figure 5 molecules-24-00591-f005:**
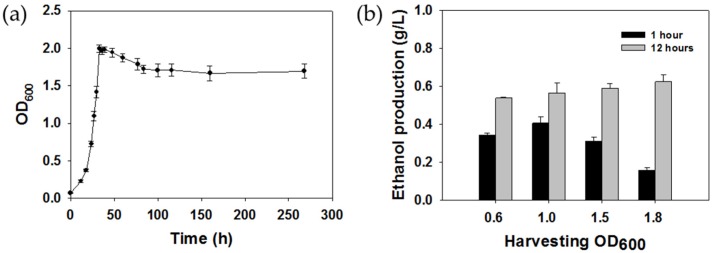
Analysis of cells in different growth phases for their ethane-to-ethanol conversion activity. Cell growth curve of *M. trichosporium* OB3b (**a**) and effect of different cells harvested in different growth phases on ethane-to-ethanol conversion (**b**).

**Figure 6 molecules-24-00591-f006:**
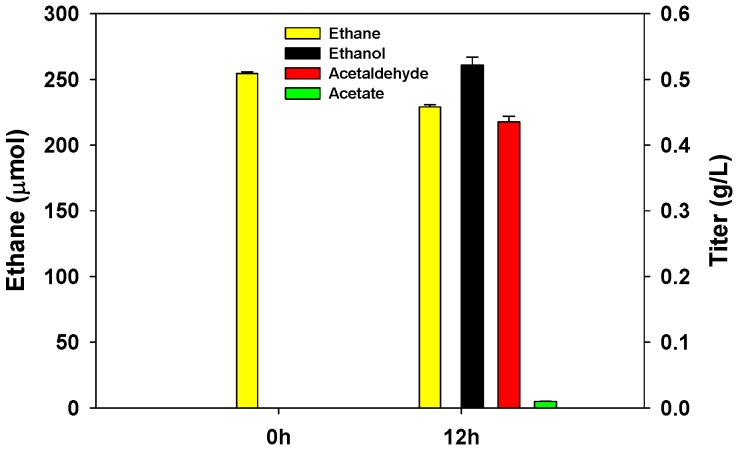
Amount of ethane in the headspace and titer of ethanol, acetaldehyde, and acetate in the reaction mixture under optimum batch reaction conditions. The reactions were carried out using 2.4 g DCW/L methanotrophic resting cell with 30% ethane in the presence of 0.3 mM EDTA and 80 mM formate.

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
