# Peer review of "Development and Optimization of the Biological Conversion of Ethane to Ethanol Using Whole-Cell Methanotrophs Possessing Methane Monooxygenase"

_molecules, 2019, doi:10.3390/molecules24030591_

Round 1
Reviewer 1 Report
The authors have optimized the biological conversion of ethane to ethanol with some representative methanotrophs.
Many studies on methane/methanol conversion using methanotrophs have already been reported but the conversion of ethane to ethanol is very interesting in the point of ethanol production. So I recommend publication of this paper in MDPI after some revisions.
Are there any difference between some representative methanotrophs mentioned in this paper, especially on the point of type I and type II?
Please comment the difference between the data of Figure 3 in ref26. In the ref26, addition of HCOONa result in the gradual decrease, while presented data shows rapid decrease in the ethanol production.
Please add the experimental methods of acetaldehyde and acetone determination.
I recommend the addition of ethane oxidation using cytochrome P450 using decoy molecule presented by Prof. Shoji et al.
Author Response
1. Thank you for your kind comment. Type I and Type II methanotrophs are distinguished by their metabolism and physiological characteristics. Therefore, there are some difference between two type of methanotrophs as ethanol production biocatalyst. In this study, we tried to evaluate various methanotroph for conversion of ethane to ethanol to find out better biocatalyst.
2. Conversion of alkane to alcohol using methanotroph has been developed by several research groups. In this study, we got more then 0.5 g/L ethanol during optimization on formate, while ref 26 got only approximately 0.15 g/L ethanol. There are different patterns and optimum condition due to selection of different strain, absence of well established method, and lack of knowledge about methanotrophic physiology. Therefore, further study is needed to elucidate exact mechanism on the effect of sodium formate on bioconversion of alkane in order to explain those gap of results.
3. Acetaldehyde and acetate were detected by HPLC with Aminex HPX-87H column and UV detector. We added this information in manuscript.
4. We added that reference.

Reviewer 2 Report
In this study, the authors have compared some representative methanotrophs in order to evaluate their ability o bioconversion of ethane to ethanol, and investigated the optimal conditions for obtaining high titer of ethanol using various concentrations of Methanol dehydrogenase inhibitor and formate in quantitative way, as well as the concentration of the cell, substrate and pH values were optimized.
Although the study analyzed several factors that may affect the ethanol production process from ethane, I believe that the effect of temperature and pressure in the process was not evaluated. I hope that in later studies this proposal is considered.
Author Response
Thank you for your kind comments.
We are considering optimization of pressure and temperature effect in further study.
